# Fermi surface chirality induced in a TaSe$_2$ monosheet formed by a Ta/Bi$_2$Se$_3$ interface reaction

Andrey Polyakov[1], Katayoon Mohseni[1], Roberto Felici [2], Christian Tusche [3,4], Ying-Jun Chen [3,4], Vitaly Feyer [3,4], Jochen Geck[5,6], Tobias Ritschel [5], Arthur Ernst [7], Juan Rubio-Zuazo[8], German R. Castro[8], Holger L. Meyerheim [1✉] & Stuart S. P. Parkin [1]

Spin-momentum locking in topological insulators and materials with Rashba-type interactions is an extremely attractive feature for novel spintronic devices and is therefore under intense investigation. Significant efforts are underway to identify new material systems with spin-momentum locking, but also to create heterostructures with new spintronic functionalities. In the present study we address both subjects and investigate a van der Waals-type hetero-structure consisting of the topological insulator Bi$_2$Se$_3$ and a single Se-Ta-Se triple-layer (TL) of H-type TaSe$_2$ grown by a method which exploits an interface reaction between the adsorbed metal and selenium. We then show, using surface x-ray diffraction, that the symmetry of the TaSe$_2$-like TL is reduced from D$_{3h}$ to C$_{3v}$ resulting from a vertical atomic shift of the tantalum atom. Spin- and momentum-resolved photoemission indicates that, owing to the symmetry lowering, the states at the Fermi surface acquire an in-plane spin component forming a surface contour with a helical Rashba-like spin texture, which is coupled to the Dirac cone of the substrate. Our approach provides a route to realize chiral two-dimensional electron systems via interface engineering in van der Waals epitaxy that do not exist in the corresponding bulk materials.

[1] Max-Planck-Institut für Mikrostukturphysik, Weinberg 2, 06120 Halle, Germany. [2] Consiglio Nazionale delle Ricerche - SPIN, Via del Politecnico, 1, Roma 00133, Italy. [3] Forschungszentrum Jülich GmbH, Peter Grünberg Institut (PGI-6), 52425 Jülich, Germany. [4] Fakultät für Physik, Universität Duisburg-Essen, 47057 Duisburg, Germany. [5] Institut für Festkörper- und Materialphysik, Technische Universität Dresden, 01062 Dresden, Germany. [6] Würzburg-Dresden Cluster of Excellence ct.qmat, Technische Universität Dresden, 01062 Dresden, Germany. [7] Institut für Theoretische Physik, Johannes Kepler Universität, A 4040 Linz, Austria. [8] SpLine, Spanish CRG BM25 Beamline at the ESRF (The European Synchrotron), F-38000 Grenoble, France. ✉email: holger.meyerheim@mpi-halle.mpg.de

Two-dimensional (2D) van der Waals (vdW) materials have emerged as fascinating materials in many fields of condensed matter physics such as e.g., in topology and magnetism. Of special interest is the vdW epitaxy where the systems exhibit a vdW gap at the interface. The adsorbate can be grown on the substrate with high structural quality and without the need for lattice-matching[1,2]. In topological materials, novel functionalities involve the locking of the electron's spin and momentum as realized in Topological Insulators (TIs), as well as in Dirac and Weyl semimetals. The chiral topologic surface state (TSS) in TIs has been found to be very effective in converting a charge current into a spin current which can exert large spin-orbit torques (SOT) in an adjacent ferromagnetic (FM) layer[3–6]. A critical issue is that the SOT efficiency resulting from the TSS can be influenced by several factors, such as the presence of bulk states, and the band-bending induced appearance of a two-dimensional electron gas, which recently has been shown to be minimized by reducing the $Bi_2Se_3$ film thickness[7]. Similarly, transition metal dichalcogenides (TMDCs) with a non-trivial electronic structure containing a heavy metal such as Mo, W, Pt, and Pd also have found remarkable interest as spin-source materials from significant charge to spin conversion[8–11].

By contrast, the metallic TMDC $TaSe_2$ has a trivial electronic structure in its bulk form and crystallizes in the 2H structure (trigonal-prismatic coordination around tantalum by selenium). The SOC lifts the spin-degeneracy of the bands inducing a spin-polarization that pins the electron spins to the out of plane direction. This scenario is referred to as an "Ising-SOC" in analogy to the "Ising-model" related to a one-dimensional chain of spins with up and down orientation only[12–17].

Here we demonstrate that in a van der vdW type heterostructure consisting of a single Se-Ta-Se triple-layer (TL) on the (0001) surface of the TI $Bi_2Se_3$ a chirality is created in the Fermi surface (FS) electronic states which couples to that of the Dirac cone across the vdW interface. The $TaSe_2$ monosheet is prepared using a simple procedure, which does not rely on exfoliation or molecular beam epitaxy methods that have been used in many previous studies but, rather, uses an interface reaction between tantalum atoms that are directly deposited onto a $Bi_2Se_3$(0001) substrate. We find that this simple method leads to flat islands that are formed from monosheets of well-ordered $TaSe_2$ with an H-type structure. The absence of the inversion center in the monosheet, in combination with the strong spin-orbit coupling (SOC) that is inherent to $TaSe_2$, results in a spin-splitting of the electronic states at the Fermi energy with oppositely spin-polarized states at the non-symmetry related K and K' points in the Brillouin zone (BZ). Until now, it has been generally assumed that monosheets of TMDC's are bulk-like. Surface X-ray diffraction (SXRD) analysis provides not only clear evidence that a single monosheet of H-type $TaSe_2$ is formed but also that the central tantalum atom in the prismatic selenium environment is vertically shifted thereby lifting the horizontal mirror plane and lowering the point group symmetry from $D_{3h}$ to the $C_{3v}$. We then use spin- and momentum-resolved photoemission spectroscopy, in combination with ab-initio calculations, to study the effect of the structural relaxation on the electronic structure. We find a very important consequence is that the spin-polarized states at the FS acquire an in-plane spin component, thereby, creating a chirality. Such a low-symmetry monosheet may serve as an efficient spin-source material that not only avoids difficulties encountered by bulk and free-electron states in $Bi_2Se_3$ but also gives rise to a more sophisticated out of plane SOT to manipulate perpendicularly magnetized ferromagnetic films, as recently demonstrated in a $WTe_2$/permalloy heterostructure[9].

## Results and discussion

**Structure analysis.** The H-$TaSe_2$ monosheet was prepared by depositing a sub-monolayer quantity of tantalum on a pristine

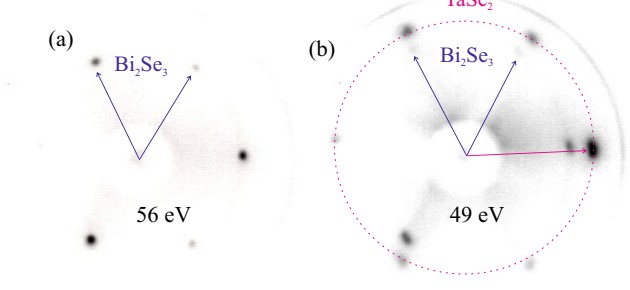

**Fig. 1 Low energy electron diffraction characterization of the H-TaSe₂ monosheet on Bi₂Se₃(0001).** LEED patterns collected for the pristine (**a**) and the H-TaSe₂ covered Bi₂Se₃(0001) surface (**b**). The outer ring of reflections belongs to the H-TaSe₂ monosheet. In combination with the SXRD experiments, the point group symmetry of the LEED pattern is determined to be 3m for Bi₂Se₃ and 6 mm for TaSe₂, the latter being related to the presence of two mutually 60 degrees rotated domains.

surface of a $Bi_2Se_3$(0001) single crystal, followed by annealing at 480 °C for several minutes. Figure 1a, b shows low energy electron diffraction (LEED) patterns of pristine and $TaSe_2$ covered $Bi_2Se_3$(0001). The pristine $Bi_2Se_3$(0001) surface exhibits a three-fold symmetric LEED pattern indicating the absence of crystal twinning. After preparation of the H-$TaSe_2$ monosheet, the substrate spots are attenuated and new spots appear that are related to the formation of a $TaSe_2$ monosheet (Fig. 1b). The in-plane lattice parameter is derived from the reflection position at approximately 1.19 reciprocal lattice units (r.l.u.) relative to the first order $Bi_2Se_3$ substrate reflections. The spot position, as well as the point group symmetry of the diffraction pattern (3 m for $Bi_2Se_3$ and 6 mm for $TaSe_2$, is also confirmed by the SXRD experiments (see Supplementary Information, section A and Fig. 3, respectively.) We find $a_0 = b_0 = 348$ pm, which corresponds to a tensile strain of 1.5% as compared to that of bulk 2H-$TaSe_2$ (343 pm)[18].

The tantalum-covered surface was studied by scanning tunneling microscopy (STM), as outlined in Fig. 2a, b. The $Bi_2Se_3$(0001) surface is characterized by terraces several hundred nanometers wide, which are separated by steps, 950 pm in height, that correspond well with the thickness of a single quintuple layer (QL) of $Bi_2Se_3$. The structure of $Bi_2Se_3$ is composed of QLs, each consisting of Se-Bi-Se-Bi-Se layers, that are separated by a van-der-Waals (vdW) gap. The QLs themselves are stacked in an A-B-C-A... sequence. In the constant current STM image in Fig. 2a ($U = -1$ V, $I = 100$ pA) $TaSe_2$ islands on the terraces appear as bright elevations with an apparent height of approximately 600 pm. The profile along the white line is given in Fig. 2b. It reflects the 950 pm high QL steps and the 600 pm high islands. Based on the STM images, the islands can be attributed to a monosheet of $TaSe_2$, whose height is expected to be approximately equal to 600 pm, i.e., half the unit cell lattice parameter $c_0 = 1.271$ nm of bulk 2H-$TaSe_2$[18]. From these observations it can be concluded that the interface between the $TaSe_2$ triple layer and the first $Bi_2Se_3$ QL is characterized by the Se-Ta-Se/Se-Bi-Se-Bi-Se layer sequence, i.e., the interface is vdW-like. This conclusion is supported by the lattice parameters of the film being close to that of bulk $TaSe_2$ with an incommensurate film to substrate relationship and by the observation that sample annealing beyond 480 °C leads to the evaporation of the film. Based on the STM image alone, no unambiguous assignment of the film structure to H- or T-type can be made, since the height of the polyhedron is nearly identical in both polytypes[19].

Detailed structural characterization was carried out by SXRD at the beamline BM25b of the European Synchrotron Radiation Facility (ESRF) in Grenoble (France) using a six-circle UHV diffractometer. In Fig. 3 the experimental intensities, $I(hk\ell)$, along several rods in reciprocal space are plotted on a log-scale versus momentum transfer ($q_z$) normal to the sample surface. In total, 86 symmetry-independent reflection intensities were collected. In contrast to bulk crystals, the coordinate $\ell = q_z/c^\star$ in reciprocal lattice is a continuous (non-integer) parameter owing to the missing lattice periodicity along the c-axis of the monosheet. The reciprocal lattice unit refers to the the $Bi_2Se_3$ substrate where $c^\star = 1/c_0 = 1/(2.864\ nm) = 0.349\ nm^{-1}$. A continuously varying intensity distribution is observed, reflecting the presence of an ultra-thin structure along the surface normal. The wide "bell-shaped" intensity profile, which is observed along all rods can be viewed as a finite size broadened Bragg reflection from the monosheet.

Quantitative analysis was carried out by fitting the experimental intensities to those observed. Owing to the high symmetry of the two-dimensional crystal structure, which belongs to the plane group $p3m1$, the structure analysis is straightforward. All atoms are located at high symmetry positions as follows: Se at (0, 0, 0) [Wyckoff position (1a)], Ta at (1/3, 2/3, z) [Wyckoff position (1b)][20] and the top-layer Se atom at (0, 0, z). Thus, the only free positional parameters are the z-positions of the tantalum and the selenium atom at the top of the Se-Ta-Se triple layer. In addition, an overall scale factor and a Debye-parameter ($B = 8\pi < u > ^2$), reflecting thermal and static disorder[21], were allowed to vary.

Solid lines in Fig. 3 represent the calculated intensities based on the structure model sketched in Fig. 4a. The fit quality is measured by the Goodness of Fit (GOF) parameter and the unweighted residuum (Ru) ($R_U = \sum ||I_{obs}| - |I_{calc}||/\sum |I_{obs}|$. Here, $I_{obs}$, $I_{calc}$ are the experimental and calculated intensities, respectively. The summation runs over all relfections. The GOF is given by: $GOF = \sqrt{1/(N-P) \cdot \sum[(I_{obs} - I_{calc})^2/\sigma^2]}$, where the difference between observed and calculated intensities is normalized to the uncertainties expressed by the standard deviation ($\sigma$) and to $(N-P)$, i.e., the difference between the number of independent reflections (N) and the number of parameters (P) which are varied. We derive values of GOF = 1.56 and Ru = 0.13. These values are very satisfactory. We note that the simulation also takes into account the presence of two rotational domains of the $TaSe_2$ unit cell with respect to the trigonal substrate surface. This is done by calculating the incoherent average of the structure factor magnitudes related to each of the two mutually 60° oriented domains. The twinning of the film structure is also the

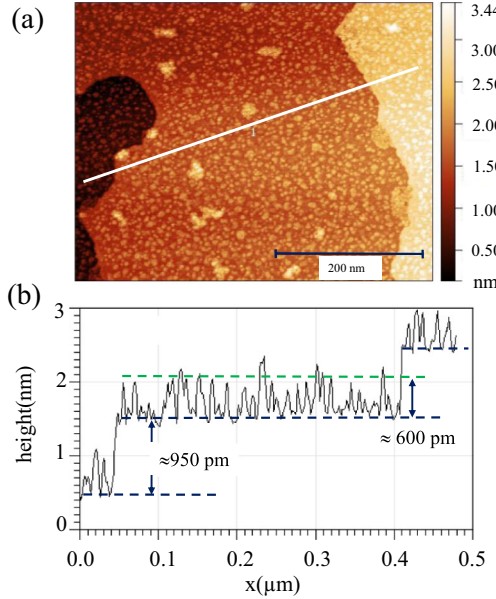

**Fig. 2 Scanning tunneling microscopy (STM) characterization of TaSe₂ monosheet on Bi₂Se₃(0001). a** Scanning tunnelling microscopy ($U = -1\,V$, $I = 100\,pA$) image of $TaSe_2$ islands (bright) on the $Bi_2Se_3$(0001) surface. **b** Height profile along the white line in (**a**). Note, the step height of the terrace (≈950 pm) related to a full QL. The height of the islands is approximately equal to 600 pm related to the height of a single Se-Ta-Se monosheet on the $Bi_2Se_3$ surface.

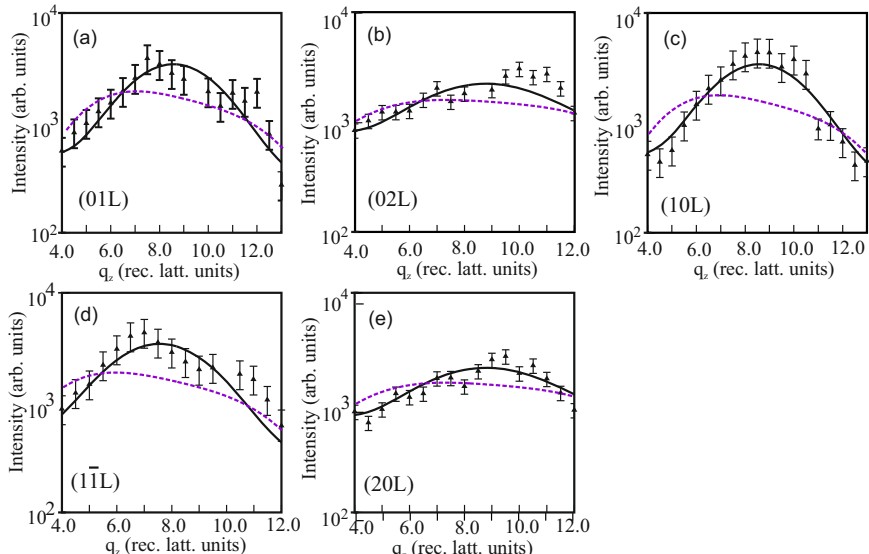

**Fig. 3 X-ray diffraction data and fit of TaSe₂ monosheet on Bi₂Se₃(0001). a–e** Experimental (symbols) and calculated (solid lines) intensities for the H-type monosheet of $TaSe_2$ on $Bi_2Se_3$(0001) along several symmetry-independent rods in reciprocal lattice as labeled. Dashed lines represent calculated intensities for the 1T-monosheet based on the same z-positions for the atoms as in the H-type. The unit of the perpendicular momentum transfer ($q_z$) is referred to the $Bi_2Se_3$ substrate lattice ($1/c_0 = 0.349\ nm^{-1}$). The 1T model can be clearly excluded based on the considerably worse fit. Error bars represent $1\sigma$ uncertainties of the experimental intensities.

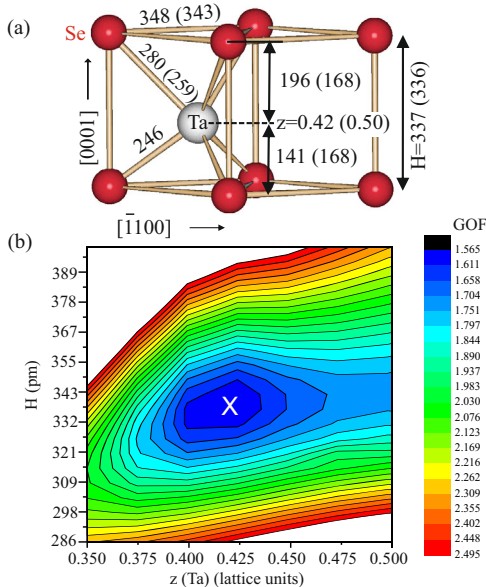

**Fig. 4 Results of the surface X-ray diffraction (SXRD) structure analysis.**
**a** Structure model of the H-TaSe$_2$ monosheet derived from the SXRD analysis. Red and gray spheres represent selenium and tantalum, respectively. Numbers indicate distances in picometer (pm) units with those in brackets referring to the bulk crystal. **b** Contour plot of GOF versus $H$ (height of the prismatic monosheet) and $z$ (relative tantalum position within the prismatic unit. The global minimum is given by the cross at $H = 337$ pm and $z = 0.42$. One contour level corresponds to a step in GOF by 3%. Uncertainties of $H$ and $z$ are estimated on the basis of $\Delta\text{GOF} = +3\%$. For details see text.

reason why the LEED pattern of the film exhibits a sixfold rather than a threefold symmetry. Figure 4a shows a schematic diagram of the structure derived from the optimum fit.

Small (red) and large (gray) balls represent selenium and tantalum atoms, respectively. The H-type monosheet is only one TL thick and stacked as in the bulk. Numbers indicate distances in picometers (pm) with those in brackets referring to bulk values in 2H-TaSe$_2$[18]. We also tested different structure models, such as the T-type monosheet structure with an octahedral selenium environment around the tantalum atom. The best fit obtained for this model is shown by the dashed lines in Fig. 3. It is clearly evident that the calculated intensities for this model fit the experimental data with considerably less accuracy than those calculated for the H-type structure. Quantitatively, Ru, as well as GOF, achieved for the T-structure are roughly a factor two larger than for the H-structure. This result unambiguously proves that the monosheet is of H-type. Similarly, the presence of the 2H-type structure can be excluded: for details, we refer to section A of the Supplementray Information.

The most important result of the SXRD analysis is that the central tantalum atom is not located in the center of the prism at $z = 0.5$ lattice units (l.u.) but is rather shifted vertically to $z = 0.42$ l.u. Simultaneously, the height of the TL is slightly increased from 336 pm in the bulk to 337 pm in the monosheet (see numbers in brackets in Fig. 4a). The downward shift of the tantalum atom by about 27 pm out of the center position involves a modification of the Ta-Se interatomic distance by about 5 and 8%. The nearest Ta-Se distance to the upper and lower selenium atom, which in bulk 2H-TaSe$_2$ structure is equal to 259 pm, is modified to 280 and 246 pm, respectively. This relatively strong vertical relaxation of the tantalum atom is attributed to the two-dimensionality and polarity of the structure, where the TaSe$_2$

monosheet experiences a strongly asymmetric environment along $z$. It is characterized by the presence of the Bi$_2$Se$_3$ substrate below and the vacuum above. This asymmetry of the structural environment involves a redistribution of charge making the $z = 0.5$ position energetically unstable. The experimental result is confirmed by ab-initio calculations (see section C of the Supplementary Information).

We have carefully evaluated the uncertainty of the distance determination as shown in Fig. 4b where the GOF is plotted versus the height ($H$) of the TL and the position ($z$) of the tantalum atom. The cross marks the global minimum at $H = 337$ pm and $z = 0.42$. Each contour level represents a step of 3% in GOF with respect to the previous level. The uncertainty of $H$ and $z$ can be estimated by the variation of the GOF upon the variation of $H$ and $z$. An increase of the GOF by 3% beyond the minimum is rated as significant, especially as the basic structure is unaffected by the very small modifications of $H$ and $z$ and because of the large number of independent reflections (86) relative to the number of free parameters (only two positional and one ADP). Under this assumption, we find uncertainties of $\Delta H = \pm 5$ pm and $\Delta z = \pm 0.02$ lattice units, the latter corresponding to $\pm 7$ pm. As a consequence of the atomic shift, the point group symmetry of the H-type prism is lowered from D$_{3h}$ ($\overline{6}$2m) to C$_{3v}$ (3m), which induces an in-plane component of the spin-polarization, as will be shown by spin- and momentum resolved photoemission experiments and by ab-initio calculations in the following. The results of the SXRD analysis serve as input for spin-resolved band structure analysis.

**Electronic structure**. Spin- and momentum resolved photoemission experiments using a spin-resolving momentum microscope[22] were carried out at the NanoESCA beamline[23] of the Elettra Synchrotron in Trieste (Italy). The sample was kept at 80 K and p-polarized light with a photon energy of $h\nu = 40$ eV were used. The incident photon beam lies in the $k_y - k_z$ plane at an angle of 25° above the surface plane. By utilizing spin-resolving momentum microscopy, a wide acceptance angle of photoelectrons of up to ±90° can be simultaneously collected. The measured momentum discs provide a comprehensive access to the spin-resolved electronic states in two-dimensional ($k_x$, $k_y$) reciprocal-space momentum maps of the photoemission intensity throughout the entire surface Brillouin zone (SBZ). Furthermore, by scanning the binding energy $E_B$, complete three-dimensional $E_B(k_x, k_y)$ maps, containing band dispersion along all directions of the SBZ, can be obtained[24].

Figure 5a, b shows the spin-averaged band structure of the pristine and the H-TaSe$_2$ covered Bi$_2$Se$_3$(0001) surface along directions connecting high-symmetry points of the first SBZ. The energy range lies between $E_F$ and 2.0 eV binding energy ($E_B$). The experimental band structure can be viewed as a superposition of two individual parts, namely that of the Bi$_2$Se$_3$(0001) substrate and that of the H-TaSe$_2$ monosheet as the hybridization between these states is low (see calculation discussed in the Supplementary Information, section C). In the vicinity of the $\overline{\Gamma}$ point the topological surface state (TSS) of the Bi$_2$Se$_3$ substrate is observed, which is commonly referred to as the "Dirac cone" with the Dirac point ($D_P$) being located several hundred meV below $E_F$. This position below $E_F$ is related to n-doping due to selenium defects. The TSS is stable upon formation of the H-TaSe$_2$ monosheet, only $D_P$ shifts slightly to higher binding energy.

The observed band structure of H-TaSe$_2$ monosheet is similar to that of bulk 2H-TaSe$_2$, qualitatively verifying that the Se-Ta-Se TL is indeed of the H-type. However, there are some differences, for instance, the number of bands crossing $E_F$ as discussed in several previous studies[19,25,26]. The H-TaSe$_2$ monosheet is

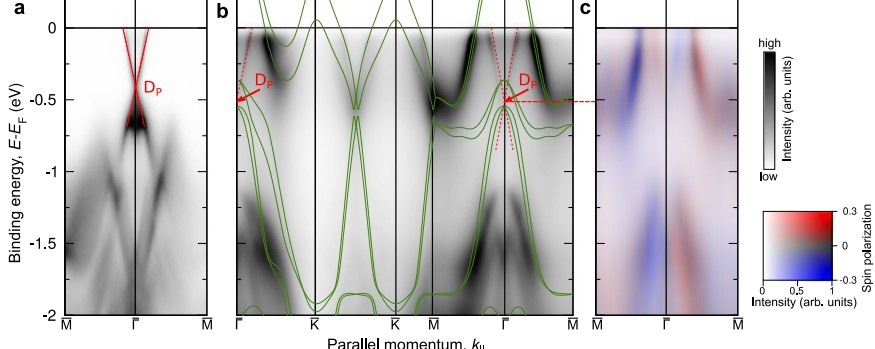

**Fig. 5 Photoemission and bandstructure of TaSe$_2$/Bi$_2$Se$_3$(0001).** Experimental spin-averaged band structure of the pristine (**a**) and the H-TaSe$_2$ covered Bi$_2$Se$_3$(0001) surface (**b**) along directions connecting high-symmetry points in the first BZ. $D_P$ denotes the Dirac point. Green lines in (**b**) represent the calculation based on the SXRD-derived structure model. The spin-resolved band structure of the H-TaSe$_2$ covered surface is shown in (**c**). The spectral density and spin-polarization projected along the y-axis are represented by the scale bars on the right.

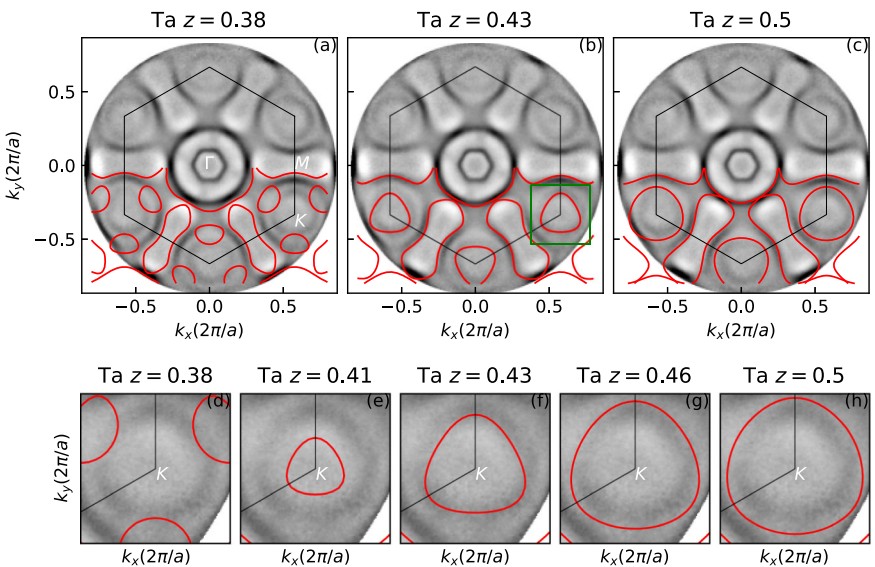

**Fig. 6 Photoemission momentum map compared with calculations. a–c** Comparison between the experimental photoemission momentum map of H-TaSe$_2$/Bi$_2$Se$_3$ at the Fermi Surface with calculated ones (red lines) for positions $z = 0.38$, $0.43$, and $0.50$ of the tantalum atom within the Se-Ta-Se triple layer. The white hexagon indicates the first Brilluoin zone. **d–h** close up of the hole pocket at the $\overline{K}$-point emphasizing the dependence of its size and position on $z$. The best match is observed for $z$ being in the range 0.41–0.43 in agreement with the structure model derived by SXRD.

metallic: there is a Ta-5d related band which crosses $E_F$ between $\overline{\Gamma}$ and $\overline{M}$. Deeper lying and also strongly dispersive bands in the binding energy range between 1 and 2 eV are related to Se-p states[26]. The electron energy distribution curves (EDC's) along the $\overline{\Gamma}$-$\overline{M}$ and $\overline{\Gamma}$-$\overline{K}$ directions are discussed in the Supplementray Information section B (Fig. 3).

The spin-resolved band structure of the H-TaSe$_2$ covered sample shown in Fig. 5c reveals that there is an antiparallel alignment between the spin texture of the TSS and that of the Ta-5d derived states which cross $E_F$ between the $\overline{M}$ and the $\overline{\Gamma}$ point. Calculations discussed in section C of the Supplementary Infomration and Supplementary Fig. 8 reproduce this scenario and suggest that the antiparallel orientation ís related to the minimization of the exchange correlation potential.

Figure 6a–c compares the experimental spin-integrated momentum map at the Fermi level ($E_F$) with the calculated FS within the first SBZ. The FS of the TaSe$_2$ monosheet is characterized by circular hole pockets around the $\overline{\Gamma}$- and the $\overline{K}$-point as well as the "dog-bone" like electron pocket around the

$\overline{M}$-point, the latter being a consequence of spin-orbit coupling (SOC) and the lack of inversion symmetry in the H-TaSe$_2$ monosheet[19,26]. Calculated momentum maps are superimposed on the experimental ones for $z = 0.38$, $z = 0.43$, and $z = 0.50$. Here, calculations refer to the "free-standing" H-TaSe$_2$ monosheet only, and the substrate TSS is not taken into account. As discussed in section of the Supplementary Information in more detail, this is justified by the weak hybridization between the corresponding states across the interface vdW gap.

Direct inspection clearly reveals that the details of the FS contour sensitively depends on the $z$-position of the tantalum atom. While for $z = 0.38$, the $\overline{K}$-pocket is too close to the $\overline{\Gamma}$ point, it moves outward until the vertical tantalum position reaches $z = 0.43$. At this point the experimental and the calculated FS contour fits fairly well, both with regard to size and location of the pockets. Increasing $z$ beyond 0.43 only leads to an increase of the size of the $\overline{K}$ hole pocket, rather than to a further outward shift. This is demonstrated in more detail in Fig. 6d–h which show the shift and the increase in size with increasing $z$. The optimum

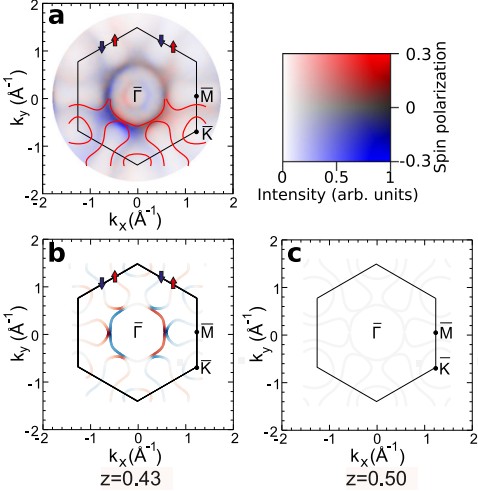

**Fig. 7 Spin-resolved photoemission momentum map.** Experimental spin-resolved photoemission momentum map of the H-TaSe$_2$ monosheet on Bi$_2$Se$_3$ at the Fermi level (**a**) compared with calculations in which the position of the tantalum atom is located at $z = 0.43$ (**b**), $z = 0.50$ (**c**). The color code quantifies the degree of the in-plane polarization along the $y$-axis as given by the scale bar on the right.

match between calculation and experiment is found for $z = 0.43$ which is in perfect agreement with the SXRD analysis.

In non-magnetic solids the generation of spin-polarized electronic states requires the breaking of the global symmetry. The transition from bulk 2H-TaSe$_2$ with its inversion symmetric P6$_3$/mmc space group (D$_{6h}$ point group symmetry) to the "unrelaxed" H-TaSe$_2$ monosheet with D$_{3h}$ point group symmetry (absence of inversion symmetry) represents such a case. The SOC lifts the spin-degeneracy of the bands inducing a spin-polarization which pins the electron spins to the out of plane direction. In analogy to the "Ising"-model, which originally refers to a one-dimensional spin-structure with up- and down-spins only, this scenario is referred to as "Ising-SOC". The Ising-SOC is regarded as being responsible for the strong enhancement of the critical field H$_{C2}$ in superconducting TMDC's such as in monosheet thick H-NbSe$_2$[12–15] and gated MoS$_2$[16,17]. Most importantly, in the H-TaSe$_2$ monosheet studied here, the point group symmetry is further lowered from D$_{3h}$, to C$_{3v}$ by the vertical shift of the tantalum atom.

In consequence of the reduced symmetry of the TaSe$_2$ monosheet a pronounced in-plane spin-polarization is observed. The spin-resolved momentum map at the FS is displayed in Fig. 7a. It shows the in-plane projected spin-polarization which is coded as parallel (red) and antiparallel (blue) to the y direction, being the same as the direction of the incident beam. The lower half of Fig. 7a shows the calculated FS contour of the H-TaSe$_2$ monosheet as a guide to the eye. The off-normal incidence of the photon beam gives rise to an intensity gradient from the top to the bottom of the momentum image due to the linear dichroism in the angular distribution of the spin-resolved intensities[27]. The spin-resolved momentum-map shows that the H-TaSe$_2$ monosheet related states at $E_F$ exhibit a pronounced in-plane spin polarization, which is a direct consequence of its low-symmetry atomic structure. In particular, the measured spin-polarization map in Fig. 7a reveals that the central circular state of the H-TaSe$_2$ FS around $\overline{\Gamma}$ exhibits a pronounced chiral spin texture. The same spin-resolved measurement also provides information on the spin-texture of the Bi$_2$Se$_3$ Dirac cone located closer to $\overline{\Gamma}$, revealing an antiparallel chirality relative to the H-TaSe$_2$ FS states (see also Supplementary Information, section C).

In order to investigate the appearance of an in-plane spin-texture in more detail we have calculated the spin-resolved states at $E_F$ using the fully relativistic Dirac-Kohn-Sham formalism as implemented in the FPLO18 package[28,29]. We carried out these calculations for different $z$-positions of the tantalum atom within the unit cell. Figure 7b, c shows the results for $z = 0.43$ and 0.50, respectively. While the latter corresonds to the "ideal" D$_{3h}$ symmetric prismatic polyhedron, the first one corresponds to the SXRD derived structure model with C$_{3v}$ point group symmetry (Fig. 4). The spin-polarization of the states is displayed as projected along the $y$-axis allowing a direct comparison with experiment. A three-dimensional image and color-coded representations of the spin polarization along all three directions are presented in Supplementary Figs. 4, 5 and 8, respectively.

In agreement with the previous discussion, there is no in-plane component of the spin texture for $z = 0.50$ (Fig. 7c). The situation changes if the tantalum atom is allowed to relax to $z = 0.43$ (Fig. 5b). In good agreement with the experimentally observed spin-polarization, an in-plane component of the spin texture appears giving rise to a chirality of the central circular state around $\overline{\Gamma}$. Chirality at the FS is observed in all states such as e.g., in the "dog-bone" like electron pocket around the $\overline{M}$-points, albeit with a weaker spectral weight. This makes the spin chirality of these states experimentally more demanding to observe. In the upper parts of Fig. 7a, b, red and blue arrows emphasize positive and negative polarization components along the "dog-bones". The weak positive and negative in-plane polarization components that are observed experimentally at these points indicate that a qualitative agreement with the calculated chiral in-plane spin-texture is found at the "dog-bone" electron pockets, in addition to the strong circular contour around $\overline{\Gamma}$. In section C (Supplementary Fig. 7) of the Supplementary Information we provide further calculations of the FS contour for the alternative structural model of the 1T-TaSe$_2$ monosheet, which, however, can be clearly ruled out by direct comparison with the experiment.

Exploiting a simple interface reaction between tantalum adsorbed on the (0001) surface of Bi$_2$Se$_3$ we have prepared a heterostructure interfacing the TSS of Bi$_2$Se$_3$ with a two-dimensional H-type TaSe$_2$-monosheet. The combination of the Dirac-type states of the Bi$_2$Se$_3$ substrate with the SOC split electronic states at $E_F$ of the symmetry reduced H-TaSe$_2$ monosheet in its proximity causes a spin-momentum locking in the latter. Our approach provides a route to realize novel chiral two-dimensional electron systems that do not exist in the corresponding bulk materials and may open new approaches in the field of spin to charge conversion and spin-orbit torques[30]. A non-magnetic TMDC in the ultra-thin film limit appears as advantageous in achieving a large SOT as it avoids the difficulties encountered with Bi$_2$Se$_3$ caused by the appearance of bulk and free electron states directly in contact with the ferromagnetic layer. Also, interesting transport phenomena may arise from the opposite chirality of the states at the $\overline{\Gamma}$-point of the adjacent films. This certainly needs to be explored in future studies. Notwithstanding these exciting possibilities for future research on this particular system, it is important to point out that the method described here can be used to prepare various selenium-based TMDC-layers on Bi$_2$Se$_3$. Our approach can therefore be used to prepare and explore various exciting materials and heterostructures.

## Methods
**Sample preparation**. The experiments were carried out in-situ under ultra-high-vacuum (UHV) conditions in different UHV chambers. The (0001) surface of the Bi$_2$Se$_3$ single crystals were cleaned by repeated cycles of mild Ar$^+$ ion sputtering followed by annealing up to 500 °C for several minutes until Auger electron spectroscopy (AES) did not show any traces of carbon and oxygen[31,32] and a highly contrasted low energy electron diffraction (LEED) pattern was observed. Tantalum

was deposited by evaporation from a tantalum rod heated by electron beam bombardment. The amount of tantalum deposited was calibrated by AES, scanning tunneling microscopy (STM) and ex-posteriori by SXRD. The H-TaSe$_2$ mono-sheet is formed by annealing the as prepared sample at 480 °C for several minutes until the LEED diffraction pattern shows extra spots related to the formation of the H-TaSe$_2$ phase.

**Surface X-ray diffraction**. The XRD experiments were carried out at the beamline BM25B of the European Synchrotron Radiation Facility (ESRF) using a six circle diffractometer operated in the $z$-axis mode. The angle of the primary beam ($\lambda = 0.71$ Å) was set to $\alpha_i = 2.0$ degrees. Integrated reflections intensities [INT$_{obs}$(hkl)] were collected by performing omega scans about the sample normal and collecting the reflected beam by using an two-dimensional pixel detector. The INT$_{obs}$(hkl) were then multiplied by instrumental correction factors $C_{corr}$ as outlined in detail in refs. [33,34] yielding the experimental intensities [$I_{obs}$(hkl)] via: $I_{obs}$(hkl)=INT$_{obs}$(hkl) × $C_{corr}$. The total uncertainty ($1\sigma$) of the $I_{obs}$(hkl) is estimated to about 10% as derived from the reproducibility of symmetry equivalent reflections.

**Spin-and momentum resolved photoemission spectroscopy**. Spin- and momentum resolved photoemission experiments using a spin-resolving momentum microscope[22] were carried out at the NanoESCA beamline[23] of the Elettra Synchrotron in Trieste (Italy). The sample was kept at 80 K, and photoelectrons from the H-TaSe$_2$ valence band region were excited by using p-polarized light and a photon energy of $h\nu = 40$ eV. Photoelectrons emitted into the complete solid angle above the sample surface were collected using a momentum microscope[35]. The momentum microscope directly forms an image of the distribution of photoelectrons as function of the lateral crystal momentum ($k_x$, $k_y$) that is recorded by an imaging detector. Here, we refer to these 2D intensity maps as "momentum discs", representing constant energy cuts through the valence band spectral function I($k_x$, $k_y$, E). For spin-resolved momentum microscopy measurements, a W(100) single crystal is introduced into the momentum microscope, such that electrons are specularly reflected at the crystal surface such that the 2D momentum image is preserved[36]. Electrons reflected from the crystal surface undergo spin-dependent scattering, such that the spin-polarization can be obtained at every point in the momentum discs over the entire SBZ[37]. The recorded intensity on the spin-resolving image detector branch depends on the electron spin being aligned parallel or antiparallel to the vertical quantization axis.

**First principle simulations**. All density functional theory calculations were performed using the FPLO18 package[28]. We modeled the Se-Ta-Se monolayer as freestanding using the experimental lattice parameters with 20 Å of vacuum in between layers in order to diminish interactions between periodic images. Brillouin zone integration was performed on a grid of $12 \times 12 \times 1$ irreducible $k$-points using the tetrahedron method. We employed the fully relativistic four-component Dirac-Kohn-Sham theory as implemented in FPLO18[29]. The generalized gradient approximation (GGA) as parametrized by Perdew et al.[38] was used to approximate the exchange-correlation potential. Spin textures represent the $k$-dependent expectation value of the spin operator.

## Data availability
The source data for all of the data and the figures in this work are available from the corresponding author upon request.

## Code availability
The source code for the calculations performed in this work is available from the corresponding authors upon request.

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

## Acknowledgements
This work is supported by the German Science foundation under Grant SPP 1666 (Topological Insulators) and by the BMBF under Grants 5K19PGA and 05K16PGB. T.R. gratefully acknowledges financial support from the German Research Foundation under Grant RI 2908/1-1. T.R. and J.G. have been supported by the Deutsche Forschungsgemeinschaft through the SFB 1143, SFB1415, and the Würzburg-Dresden Cluster of Excellence EXC 2147 (ct.qmat). The authors thank F. Weiss for technical assistance. H.L.M., A.P., K.M., and R.F. thank the staff of the ESRF for their support and hospitality during their stay in Grenoble. H.L.M., C.T, A.P. and Y.J.C. also thank the staff of Elettra for their help and hospitality during their visit in Trieste, and G. Zamborlini and M. Jugovac (PGI-6) for assistance, in using the NanoESCA beamline.

## Author contributions
S.S.P.P. and H.L.M. devised the experiments. A.P. investigated the sample preparation and characterization by STM and LEED. H.L.M., K.M., G.C., R.F., A.P., and J.R.-Z. carried out the X-ray diffraction experiments at the ESRF. Data analysis was done by K.M. and H.L.M. The momentum microscopy experiments at Elettra were carried out by C.T., V.F., H.L.M., A.P., and Y.J.C. T.R., J.G., and A.E. performed the first principle simulations. All authors contributed to the interpretation and discussion of the results. H.L.M., S.S.P.P., and C.T. wrote the manuscript with input from all authors. All authors have read and approved the manuscript.

## Funding

## Competing interests
The authors declare no competing interests.
