## [Peer Review File · Nature Communications]

REVIEWER COMMENTS

Reviewer #1 (Remarks to the Author):

This is an interesting work where the authors focused on identifying new material systems with spin-momentum locking and creating heterostructures with new spintronic functionalities. To do this, they investigated a van der Waals-type heterostructure consisting of the topological insulator Bi₂Se₃ and a single Se-Ta-Se triple-layer (TL) of H-type TaSe₂, which exploits an interface reaction between the adsorbed metal and selenium. The authors claim, that using surface x-ray diffraction, the symmetry of the TaSe₂-like TL is reduced from D_{3h} to C_{3v}, resulting from a vertical atomic shift of the tantalum atom. This work seems to be systematic; however, the experimental data do not fully show evidence:

- The ARPES experiment needs to be studied in more detail. For a deeper understanding of electronic band structure, momentum and energy distribution curves should be shown. Fig 5b) and Fig 5c) are very weak, and it is hard to see experimental bands. The bands should be seen without guiding lines from \hbar harmonic and distribution curves; the same with Fig 6. The calculated lines of the Fermi surface profiles from the photoemission momentum map are unreadable. Similar issue with the Fig. 1, where LEED picture should be shown with better contrast to notice details.
- To prove TSS's nature (2 vs 3D band character), the ARPES should be done with different photon energy.
- The authors claim: "The observed band structure of TaSe₂ monosheet is similar to that of bulk 2H-TaSe₂, verifying that the Se-Ta-Se TL is indeed of the H-type. However, there are some differences, for instance, the number of bands crossing EF as discussed in several previous studies" This part should be discussed in more details. Especially, which part of the band structure is similar and influences the number and band curvature.
- It is commonly known that simple generalized gradient approximation (GGA) can overestimate bandgaps in topological materials. Did the Author try to use different exchange-correlation functional in context, where slightly changing one atom's position can modify band structure so strong?

Reviewer #2 (Remarks to the Author):

The authors have studied the electronic structure of monolayer metallic TaSe₂ growth on the topological insulator-Bi₂Se₃ surface, using spin-ARPES measurement and density functional theory.

The authors have mentioned that due to the distortion in monolayer TaSe₂, in-plane spin-texture has emerged in the electronic structure, which might be potential for spintronics phenomenon such as spin-orbit torque. Although the spin-APES measurement looks nice, the analysis of the results is not convincingly discussed as well as analyzed in support to the experimental findings. The content of the manuscript is contemporary and important for future nanoscale spintronic research, however, I think that manuscript in the present form is not suitable and is not novel enough. Therefore, I can not recommend the manuscript to be published in Nature communication. Few criticism are listed below:

1) It is mentioned that monolayer 2H-TaSe₂ has space group $P3m1(164)$ which has inversion symmetry, rather it will be $P6m2(187)$, which has broken space inversion symmetry.

2) One of the main finding of the manuscript is the vertical distortions of Ta atoms in TaSe₂ layer, however, there is not any discussion and analysis about the mechanism behind a large distortion $\sim 0.27 \text{ \AA}$ ($\sim 16\%$). Such large distortion is unlike features in vdW bonded heterostructure, therefore, it is expected to be create metastable or unstable phase, which might be stabilized by growth atmosphere. It is not clear why vertical distortion is favorable in comparison to that of in-plane one which is expected to be dynamically more stable. Therefore, the mechanism behind the such distortion is needed to be established.

3) There is no clear explanation why 1T polymorph is not considered for the proposed electronic structure as both 1H and 1T phase of TaSe₂ are energetically close. In some extend, the electronic structure of 2H phase has some features in close relation with the Fermi surface plot. However, the electronic structure of distorted 1T also need to be discussed, as the spin-orbit splitting is quite different in distorted 1T and 1H phase. These analysis will be useful as the spin-orbit splitting at the K point (Fig.6a,b,c) is not evident in the Fermi surface.

4) The authors only compare the spin-texture of distorted monolayer of 1H-TaSe₂ with ARPES fermi surface. However, the source of spin-texture could be induced by the proximity of Bi₂Se₃. Therefore, interface related spin-orbit coupling is necessary to be discussed. Even though equilibrium spin-texture is out-of-plane in 1H-TaSe₂, the in-pane non-equilibrium spin-texture can be induced at the interface and this non-equilibrium spin-texture is mainly important for spin-orbit torque (Nano Lett. 2020, 20, 4, 2288–2295).

5) The author did not discussed the origin of inner electron pocket around Γ point in the Fermi surface (Fig.6), which might be related the topological surface states of Bi₂Se₃, responsible for inducing in-plane spin-texture, even if pristine 1H-TaSe₂ does have only out-of-plane spin-texture.

Such inner pocket may be also related to hybrid 1H-1T phase of TaSe₂ (J. Phys. Chem. C 2021, 125, 1, 1150–1156).

6) The author used a fancy/unusual phrase "Ising-SOC", which is need to be explained if there is any connection to "Ising" picture.

7) The introduction is addressed to the very specific community of the spintronics researcher, although it is expected to attract to the researcher broadly.

8) Typos: A) In the caption of fig. (c)-(h)  (d)-(h); B) To be consistent : angle-resolved  momentum resolved

Reviewer #3 (Remarks to the Author):

The manuscript of Polyakov et al. describes a convenient method to make a novel topological heterostructure. Through detailed surface X-ray diffraction and spin- and angle- resolved photoemission spectroscopy measurements, the authors conclude that the system acquires an in-plane spin-polarization due to the structural symmetry breaking along the surface-normal direction. I think the work is interesting and potentially it can be published in Nature Communications. However, the manuscript should be more convincing, so I would like the authors to address the following comments.

1. The authors write that "A non-magnetic TMDC in the ultra-thin film limit appears as advantageous in achieving a large SOT as it avoids the difficulties encountered with Bi₂Se₃ caused by the appearance of bulk and free electron states directly in contact with the ferromagnetic layer." However, as I can see in Fig. 5, Bi₂Se₃ gets more electron doped by the heterostructure formation. Since TaSe₂ is only one unit-cell thick, the bulk states of Bi₂Se₃ should surely contribute to the transport in this system. Is this really an advantage of this system?

2. In Fig. 1, I can see a ring in the LEED pattern for pure Bi₂Se₃ as well as for the TaSe₂ covered Bi₂Se₃. What is the origin of this? Furthermore, I do not understand the figure caption: "3m for Bi₂Se₃ and p6mm for TaSe₂". How can this be concluded from this data? Fig. 1(b) does not seem to be six-fold symmetric to me.

3. Is the Dirac cone of Bi₂Se₃ visible in Fig. 5 due to the fact that the surface is not fully covered with TaSe₂? Or is it still visible even after the surface is fully covered? It has been known that only a deposition of a monolayer can significantly alter the Dirac-cone dispersion (refer to PRL 107, 166801 for example) and I am wondering why the original Dirac cone is observed so clearly.

4. I have checked Ref. 15 and the results of the spin-resolved band dispersion for the Au(111) surface states are nicely presented. However, the data taken with the same apparatus for the present sample seems to be a bit confusing.

First of all, what is the value of the P_{\max} and P_{\min} in the spin-polarization?

Secondly, I do not think the data shown in Fig. 7(a) is antisymmetric with respect to $k_x=0$ (the horizontal axis should be k_x , not just k), although the tendency can be seen for the central hexagon. However, for the bone-like contour, it is quite questionable if the experimental data and the calculation are consistent. What is the origin of this?

Thirdly, the features of the electron pocket at the K point are hardly visible in Fig. 7(a). Can the author provide figures like Fig. 6(d)-(h) concerning the spin-polarization?

5. I would love to see the situation for spin-polarization in the other directions, especially the out-of-spin component. Although the experimental data may only be for the y -direction, it should be possible to show them in the calculation.

In conclusion, I think the structural analysis shown in this manuscript is very nice and convincing. However, the spin-ARPES part as well as its comparison with the *ab initio* calculation should be brushed up for publication in Nature Communications.

REVIEWER COMMENTS

The authors want to thank the Referees for carefully reading our manuscript and their valuable comments and suggestions. As far as we understand their reports our study is rated as "interesting", "systematic", "nice" as well as "potentially publishable" in Nature Communication. The Referees raise a number of important questions and suggestions which we have thoroughly considered. Following the Referee's comments we have substantially revised the manuscript.

Most importantly we have carried out new calculations which are discussed in the main text as well as in the newly added Supplementary Information (SI). Furthermore we have included new data of our XRD experiment as well as additional calculations in the main text and the SI to provide more evidence for the validity of the structure model. Also, several misunderstandings have been removed and clarifications have been made in the main text. In the following we provide a point-by-point response to the Referee's comments. Revisions in the text are indicated **by blue letters**.

We are confident that we can convince the Referees that our manuscript meets the standards of quality, validity and novelty for publication in Nature Communications.

Reviewer #1 (Remarks to the Author):

This is an interesting work where the authors focused on identifying new material systems with spin-momentum locking and creating heterostructures with new spintronic functionalities. To do this, they investigated a van der Waals-type heterostructure consisting of the topological insulator Bi₂Se₃ and a single Se-Ta-Se triple-layer (TL) of H-type TaSe₂, which exploits an interface reaction between the adsorbed metal and selenium. The authors claim, that using surface x-ray diffraction, the symmetry of the TaSe₂-like TL is reduced from D_{3h} to C_{3v}, resulting from a vertical atomic shift of the tantalum atom. This work seems to be systematic; however, the experimental data do not fully show evidence:

- The ARPES experiment needs to be studied in more detail. For a deeper understanding of electronic band structure, momentum and energy distribution curves should be shown. Fig 5b) and Fig 5c) are very weak, and it is hard to see experimental bands. The bands should be seen without guiding lines from Γ harmonic and distribution curves; the same with Fig 6. The calculated lines of the Fermi surface profiles from the photoemission momentum map are unreadable. Similar issue with the Fig. 1, where LEED picture should be shown with better contrast to notice details.*

REPLY:

We thank the Referee for her/his important comments. It helped us to improve the data presentation and to add new information. First we want to note that the lines in Figure (5) and (6) are not guiding lines but are rather calculated bands based on the x-ray diffraction structure model.

We have replotted Figures (5) and (6) in order to enhance the contrast of the photoemission data as well as the calculated lines which are superimposed on the data. We emphasize that the photoemission data are original ones without additional treatment such as differentiation. Furthermore, we have added the EDC's for the $\bar{\Gamma}$ - \bar{K} and the $\bar{\Gamma}$ - \bar{M} direction in Fig. 3 of the new Supplementary Information. We have also improved the LEED pattern (Fig.1 of the main text) by enhancing the contrast (see also Reply to the comment of Referee II).

- *To prove TSS's nature (2 vs 3D band character), the ARPES should be done with different photon energy.*

We thank the Referee for her/his comment. We think, that the two-dimensional character of the Topological Surface State of the Bi₂Se₃ substrate which is visible as a warped hexagon is proven and needs not to be demonstrated again in our study.

- *The authors claim: "The observed band structure of TaSe₂ monosheet is similar to that of bulk 2H-TaSe₂, verifying that the Se-Ta-Se TL is indeed of the H-type. However, there are some differences, for instance, the number of bands crossing EF as discussed in several previous studies" This part should be discussed in more details. Especially, which part of the band structure is similar and influences the number and band curvature.*

REPLY:

We thank the Referee for this advice. Following this we have added two sentences in the main text and refer to several publications [Refs.17, 24, 25] which detail the differences between the band-structure of the monosheet as compared to that of the bulk.

- *It is commonly known that simple generalized gradient approximation (GGA) can overestimate bandgaps in topological materials. Did the Author try to use different exchange-correlation functional in context, where slightly changing one atom's position can modify band structure so strong?*

REPLY:

We thank the Referee for this important comment, which helps us to remove some possible sources of misunderstanding in our manuscript. First, we would like to mention that the calculations involving the z-component of the Ta atom in Fig.5,6 and 7 did not consider the Topological Surface State of the substrate, but rather only the electronic structure of the free standing" TaSe₂ monosheet. Its band structure has no bandgap and is thus metallic in character and it is not of topological nature. We used the local spin density approximation (LSDA) which very well reproduced the experimental photoemission data (as shown in Figs. 5, 6 and 7).

To clarify this point we have added several sentences in the main text emphasizing that our calculations only refer to the free-standing TaSe₂ monosheet whose electronic structure is not topological in nature. Additionally, we repeated the calculations using PBE (GGA) exchange-correlation potential for the two different Ta z-positions $Ta_z=0.43$ and $Ta_z=0.5$ to show that the exchange-correlation potential has only a minor effect on the band structure (see the figure below). The changes resulting from the displacement of the Ta atom away from the high-symmetric central position are considerably more pronounced.

Fig 1: Band structure for different Ta z-positions [$z=0.43$ (left) and $z=0.50$ (right)] and different exchange-correlation potentials (PBE (red) and LSDA (blue)). The z-position is dominant for the character of the band structure.

Reviewer #2 (Remarks to the Author):

The authors have studied the electronic structure of monolayer metallic TaSe2 growth on the topological insulator-Bi2Se3 surface, using spin-ARPES measurement and density functional theory. The authors have mentioned that due to the distortion in monolayer TaSe2, in-plane spin-texture has emerged in the electronic structure, which might be potential for spintronics phenomenon such as spin-orbit torque. Although the spin-APPES measurement looks nice, the analysis of the results is not convincingly discussed as well as analyzed in support to the experimental findings. The content of the manuscript is contemporary and important for future nanoscale spintronic research, however, I think that manuscript in the present form is not suitable and is not novel enough. Therefore, I cannot recommend the manuscript to be published in Nature communication. Few criticism are listed below:

Reply:

We thank the Referee for her/his comments. Unfortunately, the Referee does not indicate what she/he means in detail by >> the analysis of the results is not convincingly discussed as well as analyzed in support to the experimental findings>>

We respect the Referees comments, but in response to her/his review we would like to add some notes:

Our work is a combination of several state of the art techniques which in addition to band structure calculation arrive at several novel results:

- (i) The geometric structure of the transition metal di-chalcogenide (here TaSe2) in the monosheet limit is subject to a symmetry reduction related to a shift of the central Ta atom. In our study this is the first direct evidence for such a relaxation, here based on a surface x-ray diffraction experiment.
- (ii) We find by spin- and momentum resolved photoemission experiments in combination with band structure calculations based on the x-ray diffraction structure data, that this

relaxation induces a chirality of the spin structure at the Fermi level in a topologically trivial band structure. Furthermore, our study provides first direct evidence for such a structurally driven modification of the spin texture in a non-magnetic TMD monosheet. The experimental ARPES data are directly compared with band structure calculations, which convincingly agree with the experimental data. We believe that the direct comparison of experimental data with calculations is the best way to analyze and the discussion of the results. As there is almost perfect agreement with them we are convinced that the experimental data as well as the conclusions are well supported. Following the suggestions of the Referee (below) we have carried out additional calculations to fit the ARPES and the XRD data which clearly indicate that the 1H structure with the central Ta atom relaxed is the correct model. We have added these in the main text as well as in the new Supplementary Information.

1) It is mentioned that monolayer 2H-TaSe₂ has space group $p3m1$ (164) which has inversion symmetry, rather it will be $\overline{p6m2}$ (187), which has broken space inversion symmetry.

Reply: We respectfully disagree with the Referee: Both space groups, $p3m1$ and $\overline{P6m2}$ are NOT inversion symmetric. In general, for a purely two-dimensional (2D) *periodic* system like the 1H-TaSe₂ monosheet, only one of the 17 possible two-dimensional (2D) plane groups are suited for the description of the structure. This needs to be distinguished from the point groups which are suited to describe the (local) structure of the unit cell: Here, the non-relaxed TaSe₂ monosheet **unit cell** has point group $\overline{6m2}$, where the six-fold inversion axis $\overline{6}$ is *identical* to $3/m$. The relaxation of the Ta atom within the prismatic coordination polyhedron formed by selenium atoms makes this horizontally lying mirror plane to vanish resulting in the $3m$ point group symmetry. According to the two-dimensional periodic structure of the film, the corresponding two-dimensional plane group is $p3m1$ (see International Tables for Crystallography). We have clarified this in the main text.

2) One of the main finding of the manuscript is the vertical distortions of Ta atoms in TaSe₂ layer, however, there is not any discussion and analysis about the mechanism behind a large distortion $\sim 0.27\text{\AA}$ ($\sim 16\%$). Such large distortion is unlike features in vdW bonded heterostructure, therefore, it is expected to be create metastable or unstable phase, which might be stabilized by growth atmosphere. It is not clear why vertical distortion is favorable in comparison to that of in-plane one which is expected to be dynamically more stable. Therefore, the mechanism behind the such distortion is needed to be established.

Reply: We are grateful for the Referee's comments, as this a very important point, and which we regret not to have discussed it properly in the original version of the manuscript. The reason for the observed structure and the vertical relaxation is the missing periodicity long the surface normal and the polarity of the structure, while simultaneously in in-plane 2D periodicity is maintained. Owing to the missing atomic coordination along the c-direction and the missing bonds, in general vertical structure relaxations as observed in our study are very common in in two-dimensional structures. We emphasize that the modification of the Ta-Se interatomic distances is less than 10%, however. In the manuscript we have added several sentences to clarify this point. ["These relatively strong vertical relaxation of the tantalum atom along the c-axis is attributed to the two-dimensionality and the polarity of the structure. There is no periodicity along the surface normal and the environment experienced by the tantalum atom is entirely different along the +z and -z direction."]

3) *There is no clear explanation why 1T polymorph is not considered for the proposed electronic structure as both 1H and 1T phase of TaSe₂ are energetically close. In some extent, the electronic structure of 2H phase has some features in close relation with the Fermi surface plot. However, the electronic structure of distorted 1T also need to be discussed, as the spin-orbit splitting is quite different in distorted 1T and 1H phase. These analysis will be useful as the spin-orbit splitting at the K point (Fig.6a,b,c) is not evident in the Fermi surface.*

We are grateful for this comment as it led us to demonstrate on the basis of XRD and ARPES that the structure of the TaSe₂ **monosheet** is 1H, while the 1T type can be excluded. This is unambiguously based on the X-ray diffraction experiment as well as by the calculation of the Fermi surface. In the revised manuscript we have added new calculations to demonstrate that the 1T structure can be excluded.

(1): we have added in Figure 3 of the main text the calculated intensities based on the 1T structure (octahedral coordination) which is fundamentally different from the 1H (prismatic coordination). The fit to the data is significantly worse than that based on the 1H structure. We have also added a calculation including the 2H structure in the Supplementary Information (Fig.2). As demonstrated, both simulations substantially disagree with the experimental data, indicating an unambiguous assignment of the monosheet to 1H.

(2): Also we have carried out new calculations of the Fermi Surface considering the 1T structure, which is in strong disagreement with the experimental data. The calculations of the FS for the 1T structure is discussed in the Supplementary Information in Fig.7.

4) *The authors only compare the spin-texture of distorted monolayer of 1H-TaSe₂ with ARPES fermi surface. However, the source of spin-texture could be induced by the proximity of Bi₂Se₃. Therefore, interface related spin-orbit coupling is necessary to be discussed. Even though equilibrium spin-texture is out-of-plane in 1H-TaSe₂, the in-plane non-equilibrium spin-texture can be induced at the interface and this non-equilibrium spin-texture is mainly important for spin-orbit torque (Nano Lett. 2020, 20, 4, 2288–2295).*

Reply:

We thank the Referee for this important comment. To investigate this issue in detail we have carried out additional calculations to study the coupling of the Topological Surface State with the TaSe₂ states across the interface vdW gap. These are discusses in the SI in section C and in Figure 8.

The calculation proved that the hybridization between the TSS and the TaSe₂ monosheet is weak but finite and indeed influences the chirality of the spin-texture relative to that of the Bi₂Se₃ substrate TSS. There is an anti-parallel alignment of the chirality of the between the TSS and those of the TaSe₂ bands which is in full agreement with experiment. We suggest that this anti-parallel alignment is a result of the minimization of the exchange energy at the interface (Pauli principle).

5) *The author did not discussed the origin of inner electron pocket around \Gamma point in the Fermi surface (Fig.6), which might be related the topological surface states of Bi₂Se₃, responsible for inducing in-plane spin-texture, even if pristine 1H-TaSe₂ does have only out-of-plane spin-texture. Such inner pocket may be also related to hybrid 1H-1T phase of TaSe₂ (J. Phys. Chem. C 2021, 125, 1, 1150–1156).*

Reply:

The Referee is fully right, the hexagonally warped inner electron pocket around the Gamma point is indeed the TSS (Dirac cone) of the Bi₂Se₃(0001) substrate crystal. We emphasize this in the main text of the manuscript (e.g. in the text page 8 as well discussing the shift of the Dirac point by 150 meV as a result of additional n-doping), but the Referee's comment led us to better emphasize this issue in the text as well as in the caption of Figs. 5,6 and 7. Our calculations including Bi₂Se₃ substrate (see SI, Figure 8) also clearly identify the Bi₂Se₃ TSS and the TaSe₂ band structure.

6) The author used a fancy/unusual phrase "Ising-SOC", which is need to be explained if there is any connection to "Ising" picture.

Reply: We have used the term "Ising SOC as it was used previously in a number of publications (some of them cited as Ref. 20-25 of the manuscript. The physical picture behind this phrase is that the spins of the non-relaxed TaSe₂ film (as well as in others, see the References) are pinned up/down out of plane only which corresponds to the 1D Ising (up-down) spin structure model. We have clarified this point in the introduction and added the References [Refs. 10-15] to this place.

7) The introduction is addressed to the very specific community of the spintronics researcher, although it is expected to attract to the researcher broadly

Reply: The Referee is right. Our study goes well beyond the spintronics community as the system investigated is a prototype for a two-dimensional van der Waals heteroepitaxial system which has attracted intense interest recently, especially in the context of magnetism: We have modified the introduction at several places and added two references to emphasize the importance of our study in this context also.

8) Typos:A) In the caption of fig. (c)-(h)  (d)-(h); B) To be consistent : angle-resolved  momentum resolved

Reply: The typos etc. were removed

Reviewer #3 (Remarks to the Author):

The manuscript of Polyakov et al. describes a convenient method to make a novel topological heterostructure. Through detailed surface X-ray diffraction and spin- and angle- resolved photoemission spectroscopy measurements, the authors conclude that the system acquires an in-plane spin-polarization due to the structural symmetry breaking along the surface-normal direction. I think the work is interesting and potentially it can be published in Nature Communications. However, the manuscript should be more convincing, so I would like the authors to address the following comments.

1. The authors write that "A non-magnetic TMDC in the ultra-thin film limit appears as advantageous in achieving a large SOT as it avoids the difficulties encountered with Bi₂Se₃ caused by the appearance of bulk and free electron states directly in contact with the ferromagnetic layer." However, as I can see in Fig. 5, Bi₂Se₃ gets more electron doped by the heterostructure formation. Since TaSe₂ is only one unit-cell thick, the bulk states of Bi₂Se₃ should surely contribute to the transport in this system. Is this really an advantage of this system?

Reply:

The Referee is fully right with her/his comment. This is exactly what we wanted to emphasize, but it might be misunderstood. As a source of spin-polarized electrons in an SOT junction Bi₂Se₃ is in principle a good choice, but the presence of the bulk valence states imposes many “difficulties” such as the bulk contribution of the conductance. What we wanted to emphasize is that H-TaSe₂ as such is probably a better choice as this mono-sheet avoids these difficulties which are encountered by Bi₂Se₃. In our study we used Bi₂Se₃ as a substrate just as a means to simply prepare the TaSe₂ mono-sheet by just annealing a layer of Ta deposited. However, in a more generalized picture it is conceivable that the TaSe₂ mono-sheet itself deposited on a different substrate, e.g. graphene, MoS₂, SiC etc. is useable as a source of spin-polarized electrons without the disadvantage of the bulk contributions encountered by using Bi₂Se₃. To what extent this is achievable is certainly subject to future investigations, where the TaSe₂ mono-sheet is prepared on different substrates. Our results however show that the structure relaxation and the related properties of the electronic structure are a property of the quasi free-standing mono-sheet itself. In the final paragraph of the revised manuscript we have clarified this point in the introduction to avoid this misunderstanding.

2. In Fig. 1, I can see a ring in the LEED pattern for pure Bi₂Se₃ as well as for the TaSe₂ covered Bi₂Se₃. What is the origin of this? Furthermore, I do not understand the figure caption: “3m for Bi₂Se₃ and p6mm for TaSe₂”. How can this be concluded from this data? Fig. 1(b) does not seem to be six-fold symmetric to me.

We regret the mistake in the caption which mixes plane group symmetry related to a periodic structure with the point group symmetry of a diffraction pattern. Correctly it must read “3m” and “6mm”. We have corrected this in the revised manuscript. Also, the LEED patterns are contaminated by a slightly defective screen which always exhibits the ring structure mentioned by the Referee. For scientific accuracy we used the LEED system available at the beamline and collected the LEED data for the same sample and the same preparation for which the ARPES data were collected afterwards. We agree that the 6mm symmetry of the first order spots of the H-TaSe₂ monosheet is not absolutely evident in figure 1(b). However, it is clearly evident from the X-ray diffraction experiments where the >>apparent>> sixfold symmetry is proven (see Fig.3 in the main text: here the (10) and the (01) are equivalent, although they are not by the p3m1 symmetry of the structure. This is due to the formation of two rotational domains of 3m symmetry. The domain formation is taken into account in the X-ray data analysis. In the revised manuscript we have improved the image in Figure 1(b) and revised the text to clarify the symmetry of the pattern: It is 6mm not derived by inspection of the LEED pattern but derived from the XRD data.

3. Is the Dirac cone of Bi₂Se₃ visible in Fig. 5 due to the fact that the surface is not fully covered with TaSe₂? Or is it still visible even after the surface is fully covered? It has been known that only a deposition of a monolayer can significantly alter the Dirac-cone dispersion (refer to PRL 107, 166801 for example) and I am wondering why the original Dirac cone is observed so clearly.

Reply: The Dirac cone is observable even in the case of a fully covered substrate surface. Similarly, the LEED spots of the substrate are still visible. The reason is that in the case of a mono-sheet the

film thickness is about 6.5 Å, which is roughly one $1/e$ length for electrons of kinetic energy in the 50 eV range. This is still sufficient to clearly observe the substrate contribution (band structure and geometric structure).

4. I have checked Ref. 15 and the results of the spin-resolved band dispersion for the Au(111) surface states are nicely presented. However, the data taken with the same apparatus for the present sample seems to be a bit confusing.

First of all, what is the value of the P_{max} and P_{min} in the spin-polarization?

Secondly, I do not think the data shown in Fig. 7(a) is antisymmetric with respect to $k_x=0$ (the horizontal axis should be k_x , not just k), although the tendency can be seen for the central hexagon. However, for the bone-like contour, it is quite questionable if the experimental data and the calculation are consistent. What is the origin of this?

Thirdly, the features of the electron pocket at the K point are hardly visible in Fig. 7(a). Can the author provide figures like Fig. 6(d)-(h) concerning the spin-polarization?

Reply: We thank the reviewer for pointing out the high quality of our previous results on the Au(111) surface state. In this context it should be noted that the mentioned measurements from Ref. 15 were performed on a high-quality single crystal surface. Compared to the present study of an *in-situ grown TaSe₂ monolayer* it is therefore not surprising that the measurements of Ref. 15 show considerably sharper features due to the high perfection of the metal single crystal. Photoemission measurements are generally sensitive to imperfections of the surface quality, and the fact to observe sharp bands and a clear Fermi surface contour of the TaSe₂ monolayer in Figs. 5-7 underlines the high quality of the as-grown film on the Bi₂Se₃ substrate.

We further like to thank the referee for pointing out some shortcomings in the presentation of Fig. 5 and Fig. 7. The scale of the spin polarization is -0.3 to +0.3. This information is provided in the revised figures. As well was the labeling of the k_x axis corrected in the revised Fig. 7.

The reviewer points out that the spin-polarization in Fig. 7a might not be fully anti-symmetric. Indeed, in places of weak intensities, e.g., at the mentioned bone-like features, a small shift towards spin-down (blue colors) polarization can be observed. This is in general not surprising, as the measured data is not symmetrized, but represents the true measured spin-polarization in the full surface Brillouin zone (SBZ). The mentioned deviations with respect to the theoretical prediction in Fig. 7b can be related to two effects. First, the calculation shows several states with different polarization which are located in a narrow region in k -space. In the experiment the resolution might not be sufficient to fully separate these states, but they will appear as blue/red coloured rims at a few places. Second, the off-normal incidence of the photon beam can, in general, lead to the emission of polarized electrons, shifting the observed spin-polarization. We nevertheless like to point out that the overall structure of the spin-texture in the central hexagon and in the bone-shaped features of the Fermi surface is reproduced well by the calculation using $z=0.43$, whereas no sizable in-plane polarization would be expected for $z=0.50$. We agree with the reviewer that a sequence of the evolution of the spin-texture for different z values would be useful. In the revised manuscript, the calculated spin-textures around the K-point have been included in the new Supplementary Fig. 6.

5. I would love to see the situation for spin-polarization in the other directions, especially the out-of-spin component. Although the experimental data may only be for the y -direction, it should be possible to show them in the calculation.

Reply: We thank the reviewer for suggesting to improve the visualization of the 3D k-space spin texture. Following his/her advice, we added a new figures in the SI (Figs. 4 and 5) which illustrate the full 3D structure of the k-space spin texture. We agree with the referee that these additional figures indeed improve the representation of our results and help to visualize the full complexity of this intriguing k-space spin texture.

In conclusion, I think the structural analysis shown in this manuscript is very nice and convincing. However, the spin-ARPES part as well as its comparison with the ab initio calculation should be brushed up for publication in Nature Communications.

REVIEWER COMMENTS

Reviewer #2 (Remarks to the Author):

I have gone through the revised manuscript and found that the authors have addressed all concerns raised by all the referees, point by points in detail and amended the revised manuscript accordingly. Therefore, I find that the quality and presentation of the manuscript has improved substantially. However, I still find that the mechanism of the vertical distortion of Ta atom, which is the origin of in-plane spin-texture, has not been explained convincingly, although the authors discussed this issue (with texts) in terms of the non-periodicity along z-direction and the polarity. The Ta-distortion and subsequent emergence of the spin-texture are the central messages/findings of the manuscript; therefore, I feel that more solid explanations are needed to be established either from experimental evidence of such vertical distortion or from theoretical analysis of energetics to induce such distortion (in comparison to the other possibilities) and the corresponding dynamical stability. Thereafter, the manuscript can be considered for publication in Nature communication.

Reviewer #3 (Remarks to the Author):

I think the authors have tried to address the comments raised by the three referees. However, I still think the correspondence between theory and experiment shown in Figs. 7(a) and (b) are not consistent with each other, and the authors have not made a good argument about this.

Thus I do not agree with the authors's statement "We nevertheless like to point out that the overall structure of the spin-texture in the central hexagon and in the bone-shapes features of the Fermi surface is reproduced well by the calculation using $z=0.43$." The shape of the contour itself does not seem to be the same as what is shown in Fig. 6.

I can agree that the spin-polarization may not reflect that of the initial-state due to final state and matrix element effects. However, the authors just write that "there is remarkable agreement between experiment and theory which for instance is clearly evident in the upper part of the image." Why is the consistency not good in the lower part of the image? Is it due to the geometry of the light incidence and the direction of the analyzer? The authors surely need to make some explanation about this in the experimental part. Otherwise, I suggest the authors to measure the z component which should show larger spin polarization as shown in Fig. 5 of the supplementary material.

The authors thank the Referees for their valuable comments and criticisms. We are very pleased that the Referees comments are very positive, and that only a few clarifications are requested to make our manuscript suitable for publication in Nat. Comm.

Following the Referees comments we have revised the manuscript and the Supporting Information. The modifications are as follows:

- (1) We have carried out additional calculations regarding the energetic stability of the structure depending on the vertical Ta position. This is now discussed in detail in the SI and mentioned in the main text.
- (2) We have revised Fig.7(a) to clarify the correspondence of the Fermi surface contour in comparison to Fig.6
- (3) The entire paragraph discussing the results of the spin-and momentum-resolved photoemission experiments has been revised following the comments of Referee #3)

Please note that the revisions in the main text and in the SI are highlighted in blue.

REVIEWER COMMENTS

Reviewer #2 (Remarks to the Author):

I have gone through the revised manuscript and found that the authors have addressed all concerns raised by all the referees, point by points in detail and amended the revised manuscript accordingly. Therefore, I find that the quality and presentation of the manuscript has improved substantially. However, I still find that the mechanism of the vertical distortion of Ta atom, which is the origin of in-plane spin-texture, has not been explained convincingly, although the authors discussed this issue (with texts) in terms of the non-periodicity along z-direction and the polarity. The Ta-distortion and subsequent emergence of the spin-texture are the central messages/findings of the manuscript; therefore, I feel that more solid explanations are needed to be established either from experimental evidence of such vertical distortion or from theoretical analysis of energetics to induce such distortion (in comparison to the other possibilities) and the corresponding dynamical stability. Thereafter, the manuscript can be considered for publication in Nature communication.

Reply:

We thank the Referee for her/his comments and further suggestions. We agree that the structural relaxation is the decisive finding of this study and that more detailed evidence would be helpful. To this end we have carried out further calculations using a slab structural model which includes the TaSe₂ monosheet located on the Bi₂Se₃(0001) surface. This is outlined in detail in Section C(a) of the revised SI.

This theoretical analysis very nicely confirms our experimental results that were derived from x-ray diffraction and photoemission experiments: the Ta atom is vertically shifted downwards to $z \approx 0.42$ (relative coordinate with respect to the height of the prism of 337 pm, see Fig. 4 of the main text and of the SI). At $z = 0.42$ the global energy minimum is found. The vertical relaxation is a manifestation of the fact that the TaSe₂ monosheet experiences an asymmetric (polar) environment which is characterized by the bulk Bi₂Se₃ crystal below and the vacuum above it. From photoemission data (Fig. 5 of the main text) it is known that the TaSe₂ monosheet is metallic. The structural

asymmetry involves a redistribution of charge towards the Bi₂Se₃ substrate ($-z$). This charge redistribution makes the symmetric bulk-like position at $z=0.5$ energetically unstable and an energetic stabilization is achieved by the relaxation of the Ta atom to the direction of higher charge density.

Redistribution of charge at the interface between a bulk crystal and vacuum to the direction of the bulk is a well-known phenomenon and commonly observed. One famous manifestation is the so called "Smoluchowski smoothing" of the surface charge density observed at clean metal surfaces which leads to an inward relaxation of the topmost atomic layer.

For the TaSe₂ monosheet the calculation also indicates a local energy minimum is found at $z\approx 0.59$, which is almost mirror symmetric to the $z=0.42$ position, but here the bond strength is weaker to the Se atoms which is attributed to the lower local charge density along the $+z$ direction. In conclusion the calculations confirm the experimental scenario and provides an explanation for the observed structural model.

Reviewer #3 (Remarks to the Author):

I think the authors have tried to address the comments raised by the three referees. However, I still think the correspondence between theory and experiment shown in Figs. 7(a) and (b) are not consistent with each other, and the authors have not made a good argument about this. Thus I do not agree with the author's statement "We nevertheless like to point out that the overall structure of the spin-texture in the central hexagon and in the bone-shapes features of the Fermi surface is reproduced well by the calculation using $z=0.43$." The shape of the contour itself does not seem to be the same as what is shown in Fig. 6.

REPLY: We thank the Referee for confirming that most of the previous questions have been addressed in our revised manuscript. As the only remaining point the Referee asks for further clarification of the measured spin-resolved Fermi surface (FS) contour compared to the theory in Fig. 7a and b. It is unfortunately not clear on which observations the referee bases the new concern that "*The shape of the contour itself does not seem to be the same as what is shown in Fig. 6.*" In particular both measurements show the same Fermi surface contour including the central hexagon of the TaSe₂ sheet, the dog-bone like structures and the weak surface state feature of the Bi₂Se₃ substrate. The main difference between these two measurements is that Fig. 7a also shows the measured in-plane (P_y) spin polarization component encoded by red/blue intensities. The measured spin polarization of the dog-bone features of the FS contour is considerably lower than the polarization of the central hexagon, thus these states correctly appear weaker in the spin-resolved image. Possible differences that might be observed between the non-spin-resolved FS contour in Fig. 6 and the spin-resolved measurement in Fig. 7 are thus directly related to the additional information on the in-plane spin-polarization in Fig. 7a.

In order to facilitate the comparison of the spin-resolved FS contour with the non-spin-resolved one, we have now additionally included the outline theoretical FS contour for the $z=0.43$ structure in the lower half of Fig. 7a as a guide to the eye. This is the same representation as used in Fig. 6, such that a direct correspondence between respective electron states in the FS and the shape of the contour is now possible.

Following the Referee's comments we have revised Fig. 7a and the entire paragraph concerning the discussion of the spin-and momentum resolved photoemission (including the rewording of the sentences that were criticized by the Referee) beginning with (see blue letters in the main text):

"... The lower half of Fig. 7(a) shows the calculated Fs contour of the TaSe₂ monosheet as a guide to the eye. ..."

I can agree that the spin-polarization may not reflect that of the initial-state due to final state and matrix element effects. However, the authors just write that "there is remarkable agreement between experiment and theory which for instance is clearly evident in the upper part of the image." Why is the consistency not good in the lower part of the image? Is it due to the geometry of the light incidence and the direction of the analyzer? The authors surely need to make some explanation about this in the experimental part.

REPLY:

The most important agreement between the experimental data and the theory is the clear appearance of spin-polarization in the central hexagon of the TaSe₂ FS. It needs to be mentioned that appearance of sizable spin polarization in this state is a direct result of the lowered symmetry by the vertical displacement of Ta atoms. This is confirmed by theory (Fig. 7c) where the non-relaxed $z=0.5$ case does not show any in-plane spin polarization. The Referee correctly recognizes that the off-normal incidence of the light leads to an asymmetry of the photoemission intensities between the upper and lower half of the k-space, since the light is incident along the k_y axis. This effect is well known as LDAD (linear dichroism in the angular dependence) and also has influence on the intensity of spin-polarized photoelectrons [e.g., see: Phys. Status Solidi RRL 12, 1800078 (2018)]. Here, this mainly affects the visibility of the spin polarization in the dog-bone structures, which are characterized by a lower polarization compared to the central hexagon. For completeness we discuss the polarization in the upper part of the image. In Fig. 7a this is emphasized by blue and red arrows highlighting the weak spin-down/spin-up polarization of these states.

In the revised manuscript the effect of the LDAD has been explained by:

„The off-normal incidence of the photon beam gives rise to an intensity gradient from the top to the bottom of the momentum image due to the linear dichroism in the angular distribution of the spin-resolved intensities [Phys. Status Solidi RRL 12, 1800078 (2018)]. ...“

In general, the measurement of spin-resolved band structures in a photoemission experiment is a rather challenging task which is always associated with a reduced counting statistics and resolution compared to the non-spin-resolved experiment (e.g., Fig. 6). By using the advantage of a full-field 2D spin-polarization analysis in combination with the intense photon beam of the Elettra synchrotron light source our present experiments already makes use of the most advanced approach to measure the spin polarization over the entire FS contour of the TaSe₂ sheet. For a recent review on this topic we like to refer to [Photoelectron Spectroscopy, Bulk and Surface Electronic Structures, Springer Series in Surface Sciences, Vol. 72, Chapter 11, ISBN: 978-3-030-64073-6 (2021)].

In the revised manuscript we have clarified the comparison between the experimental and theoretical spin textures:

“In particular, the measured spin-polarization map in Fig. 7(a) reveals that the central circular state the TaSe₂ FS around Γ exhibits a pronounced chiral spin texture. The same spin-resolved measurement also provides information on the spin-texture of the Bi₂Se₃ Dirac cone located closer to Γ , revealing an antiparallel chirality as compared to the H-TaSe₂ FS (see also SI section C).”

and

“... In good agreement with the experimentally observed spin-polarization, an in-plane component of the spin texture appears giving rise to a chirality of the central circular state around Γ”

and

“... The weak positive and negative in-plane polarizations components that are observe experimentally at these points indicate that a qualitative agreement with the calculated chiral in-plane spin-texture is also found at the "dog-bone" electron pockets, in addition to the strong circular contour around Γ”

Otherwise, I suggest the authors to measure the z component which should show larger spin polarization as shown in Fig. 5 of the supplementary material.

REPLY:

This is the only point where we disagree with the Referee as explained below:

While the spin-resolved measurement of the z-component would probably result in nicer images due to the larger expected polarization values, this would be a completely different experiment that we believe would not provide additional insight into the scientific case of our manuscript regarding the induced in-plane chiral spin structure in the structurally relaxed TaSe₂ monosheet.

In particular, the calculation of the P_z spin polarization in Fig. 6 of the SI shows that the z-component is not sensitive to the z-relaxation and the associated symmetry breaking of the TaSe₂ monosheet. The measurement of the z-component of the spin polarization thus would also not provide new information. By contrast only the in-plane spin-polarization vanishes for the high-symmetric position of the Ta atom at z=0.5. The in-plane polarization that is observed in Fig. 7a thus is a direct consequence of the lowered symmetry by the vertical relaxation.

It also needs to be emphasized that, albeit the P_z polarization values are larger than the in-plane components, currently no photoemission experiment exists worldwide that would allow the full-field 2D spin-analysis of the P_z component over the entire FS, comparable to what is shown in Fig 7a for the in-plane P_y component. The experiment suggested by the Referee thus is clearly beyond the current state-of-the-art, and as explained above, also would not provide additional scientific information on the symmetry lowering of the TaSe₂ monosheet.

REVIEWERS' COMMENTS

Reviewer #2 (Remarks to the Author):

In the revised manuscript, the authors have addressed my concerns about the mechanism behind the vertical Ta distortion through additional DFT calculations and corresponding experimental analysis. The analysis looks convincing and subsequently, the manuscript has been revised accordingly both in the main text and supplementary. Therefore, I find that the quality and presentation of the manuscript has improved substantially, and the central message of the manuscript has been addressed more clearer fashion. Therefore, I recommend the manuscript for publication in Nature Communication in the present form.

Reviewer #3 (Remarks to the Author):

The authors have properly taken into account my comments regarding the comparison between the experiment and theoretical analysis concerning the in-plane spin polarization and I have no further objections. I think the manuscript can be published in Nature Comm.